# High-Performance Anion Exchange Chromatography with Pulsed Amperometric Detection (HPAEC–PAD) and Chemometrics for Geographical and Floral Authentication of Honeys from Southern Italy (*Calabria region*)

**DOI:** 10.3390/foods9111625

**Published:** 2020-11-07

**Authors:** Sonia Carabetta, Rosa Di Sanzo, Luca Campone, Salvatore Fuda, Luca Rastrelli, Mariateresa Russo

**Affiliations:** 1Department of Agriculture Science, Food Chemistry, Safety and Sensoromic Laboratory (FoCuSS Lab), University of Reggio Calabria, Via dell’Università, 25, 89124 Reggio Calabria, Italy; rosa.disanzo@unirc.it (R.D.S.); salvatore.fuda@unirc.it (S.F.); mariateresa.russo@unirc.it (M.R.); 2Department of Biotechnology and Biosciences, University of Milano-Bicocca, Piazza della Scienza 2, I-20126 Milan, Italy; luca.campone@unimib.it; 3Department of Pharmacy, University of Salerno, 84084 Salerno, Italy; rastrelli@unisa.it

**Keywords:** HPAEC–PAD, HPLC–RI, honey, Calabria, principal component analysis, linear discriminant analysis

## Abstract

High-performance anion exchange chromatography with pulsed amperometric detection (HPAEC–PAD) combined with chemometric analysis was developed to describe, for the first time, the sugar profile of sixty-one honeys of different botanical origin produced in southern Italy (Calabria Region). The principal component and linear discriminant analysis used to describe the variability of sugar data were able to discriminate the honeys according to their botanical origin with a correlation index higher than 90%. For the purpose of the robustness of the conclusions of this study, the analytical advantages of the HPAEC–PAD method have been statistically demonstrated compared to the official Italian HPLC–RI method (Refractive Index detection). Finally, as the characterization of the floral and geographical origin of honey became an important issue due to high consumer demand, 13 acacia honeys originating from Europe and China were studied by using the same method. By chemometric method it was possible to discriminate the different geographical origin with an index of 100%. All results proved the possibility to identify the sugar profile obtained by HPAEC–PAD combined with a robust statistical analysis, as a tool of authentication.

## 1. Introduction

According to the European Union Council Directive 2001/110/EC and Italian Law Decree n. 179/2004, honey is “the natural sweet substance produced by Apis mellifera bees from the nectar of plants or from the secretions of living parts of plants or excretions of plant-sucking insects on the living parts of plants, which the bees collect, transform by combining with specific substances of their own, deposit, dehydrate, store and leave in honeycombs to ripen and mature” [1].

According to the Directive 2001/110/EC, Italian Law Decree n. 179/200, and Codex Alimentarius Commission (amended in 2019) [2], when placed on the market as honey or used in any product intended for human consumption, honey must meet specific composition criteria classified as essential and additional composition and quality factors. Composition and quality factors include sugars, water-insoluble content, electrical conductivity, free acid, diastase activity. Sugars are the main components of honey and for this reason they are included among the essential composition factors. In particular, glucose and fructose derive, in part, directly from the nectar of the botanical species and in part from the action of the enzymes secreted by the salivary glands of the bees on sucrose. The aforementioned regulations set clear limits: the fructose and glucose content (sum of both) in blossom honey must be not less than 60 g/100 g while sucrose content—in general not more than 5 g/100 g, not more than 10 g/100 for honey of false acacia (*Robinia pseudoacacia*), alfalfa (*Medicago sativa*), Menzies Banksia (*Banksia menziesii*), French honeysuckle (*Hedysarum*), redgum (*Eucalyptus camadulensis*), leatherwood (*Eucryphia lucida*, *Eucrphia milliganii*), *Citrus* spp., and not more than 15 g/100 g for lavender (*Lavandula* spp.), borage (*Borago officinalis*). The water-insoluble content, which allows for the evaluation of impurities in honey, in general, must not be more than 0.1 g/100 g, and the electrical conductivity, which allows the evaluation of the richness in mineral salts. This may vary greatly with the botanical species from which the honey derives, in general, not more than 0.8 mS/cm.

Among the other additional factors of the composition and quality of honey are the following: free acidity, related to the content of organic acids, which must be, in general, not more than 50 meq/kg and diastase activity (Scale Schade). Diastase is a group of enzymes including manly a-amylase which degrades starch to a mixture of the disaccharide maltose, the trisaccharide maltotriose and oligosaccharides known as dextrins. A unit of diastatic activity is defined as the amount of α-amylase, which will convert 0.01 g of starch to the prescribed end point in one hour at 40 °C. The results are expressed in Schade units per gram of honey and called Diastase Number (DN). Diastase activity must be, in general, less than 8 Schade units, and in the case of honeys with a low natural enzyme content (e.g., citrus honeys), not less than 3 Schade units.

The chemical composition can vary among different honeys due to different factors, such as the botanical origin, geographic area, season, technology used for honey extraction, and storage conditions.

Honeys can be classified according to their floral source as monofloral (or unifloral) honey or multifloral (polyfloral) honey, if arising predominantly from a single or from several plant species, respectively. In addition to the botanical origin, honey can also be classified by its geographical origin. According to the EU Directive, the countries of origin, where the honey is harvested, must be declared on the label and declared as: “blend of EU honeys” (for example a blend of honey from more than one member state), “blend of non-EU honeys” (a mix of honey from more than one country outside the EU)”, “blend of EU and non-EU honeys” (e.g., a mix of EU and non-EU honey). Furthermore, there are particular types which are those coming from specific areas within the EU, processed and prepared using recognized standards—whose chemical characteristics are well defined and related to the specific region or particular local environment with inherent natural and human factors—which must bear labels of protected designation of origin (PDO) and/or protected geographical identification (PGI) labels (Regulation (EEC) No. 2081/92).

The increasing demand for monofloral, PDO and PGI honeys, generally perceived as high-quality products, produced an increase in their commercial value and, at the same time, an increase in counterfeiting.

After China, the EU is the second honey producer in the world but also a great importer of honey, as confirmed by the widespread presence on the European and Italian market of non-EU honeys, often characterized by a low cost and poor quality [3,4,5,6].

As a result, many studies were developed aimed at evaluating the quality and authenticity of honey both to protect consumers and to combat counterfeiting and the consequent unfair competition between producers.

The main concerns related to the authenticity of honey were focused on the addition of substances (syrups or sugars), and above all, on their botanical and geographical origins [7].

Traditionally, honey has always been used for healing wounds and burns and for the treatment of colds and sore throats. Over the years, many scientific studies have demonstrated the antibacterial, hepatoprotective, hypoglycemic, antihypertensive, gastroprotective, antifungal, anti-inflammatory and antioxidant effects of the different honeys, with more or less marked action depending on the botanical and/or geographical origin [8,9,10].

In fact, the composition of honey, and thus its identity and quality, can vary not only due to factors such as botanical origin and geographical area, but also according to the harvesting season, the extraction technology, and the storage and preservation conditions [11,12].

A widely used method to confirm the botanical origin of honey is melissopalynological analysis, which identifies pollen species in honey. However, this method requires professional inspectors for visual identification [4]. Alternative methods, including high-performance liquid chromatography (HPLC) [5], gas chromatography–mass spectrometry (GC–MS) [6], physico-chemical analysis [10,11], Raman spectroscopy [13] and near-infrared spectroscopy [14], could be used.

Many authors reported that the carbohydrate composition and the percentage of individual sugars were directly related to nectar or honeydew [15,16,17]. In particular, honey sugars derive from the action of numerous enzymes on the components of the nectar. The result is a complex mixture consisting of approximately 70% monosaccharides (glucose and fructose) and 10%–15% disaccharides of glucose and fructose with a glycosidic bond in different positions and configurations [13], trisaccharides, and several larger oligosaccharides. The content of disaccharides (mainly, maltose and sucrose) were considered as a tool for the characterization and identification of honey [10,14,18,19].

In fact, minor honey sugars may be useful for the determination of floral origin and may act as a “fingerprint” for a sample’s floral source [3,16,20,21].

Besides the reducing sugars (glucose and fructose), the amount of sucrose is a very important factor for evaluating honey quality. Therefore, carbohydrate analysis is important as a honey quality parameter and for the floral origin determinations. For the quantification of the carbohydrates in honey, different research groups proposed high-performance anion-exchange chromatography with pulsed amperometric detection (HPAEC–PAD). Cordella et al. demonstrated that the HPAEC–PAD technique can be used in an automated chemometric approach for the honey characterization. [5,12,13,14,15,16,17,18,19]. In the EU, Italy is among the most important producers of honey. Italy has a long tradition in the production of high-quality honeys and thanks to the biodiversity of its territories, a large number of monofloral and multifloral honeys are produced [22,23,24].

Due to the particular environmental conditions and climate, the Calabria Region is, in Italy, among the most important production areas. Despite the importance of honey production in the Calabria food chain, to our knowledge, no organic study has ever been conducted with the aim of defining the sugar profile. Moreover, for the purpose of the robustness of the conclusions of the present study, the analytical advantages of the HPAEC–PAD method were statistically demonstrated in comparison to the HPLC–RI (High performance liquid chromatography with refractive index detection) Italian official method (Italian G.U 185/2003) [25,26].

In this work, the sugars profiling of Calabrian honeys, also compared with honey from Europe and China, were studied by high-performance anion-exchange chromatography with pulsed amperometric detection (HPAEC–PAD). Chemometric analyses were used to describe the variability of sugar data associated with the different honey profiles to be related to either geographical and floral authentication.

## 2. Materials and Methods

### 2.1. Chemicals and Reagents

Glucose, fructose, sucrose, isomaltose, nigerose, maltose, kojibiose, α-β trehalose, panose, palatinose, and erlose were obtained from Sigma Chemical Co. (St. Louis, MO, USA). Isomaltotriose was purchased from Supelco (Bellefonte, PA, USA) and melezitose from Fluka (Madrid, Spain). NaOH 5M was purchased from Labochimica (PD, Italy) and Acetonitrile HPLC grade from J.T. Baker (Italy).

### 2.2. Honey Samples

A total of 74 monofloral honey samples, divided into three different clusters, were collected for this study. Specifically, the 61 honey samples obtained directly from Italian beekeepers (Calabria Region), in 2019, refer to the first cluster, called Calabria. According to the botanical origin, these honeys were classified into: 13 citrus (*Citrus* spp.), 10 eucalyptus (*Eucaliptus globulus*), 16 chestnut (*Castanea sativa*), 10 acacia (*Robinia Pseudoacacia* L. Locust) and 12 sulla honeys (*Hedysarum Coronarium* L.). All samples were stored refrigerated at 4 °C until analysis.

The other two clusters included 13 acacia honeys purchased on markets during the same year in 2019. The second cluster, named Europe, included 7 samples (1 from Romania, 1 from Bulgaria, 2 from Hungary, 1 from Croatia and 2 declared as Europe) and in the third cluster, named China, 6 samples (1 from Jilin province, 2 from Zhejiang province and 3 from Shandong province). For these commercial samples, the botanical and geographical origin were declared on the label, while for the local honeys, the botanical origin was directly certified by beekeepers.

### 2.3. Sample Preparation

Every honey sample was accurately weighed (1.0 g), dissolved in deionized water in a 50 mL volumetric flask and the volume was properly completed. The obtained sample was stirred and 1 mL of this sample was dissolved in deionized water in a 25 mL volumetric flask and the volume was completed to obtain a final concentration of 0.8 mg/mL.

Three honey samples (2 of citrus and 1 of chestnut, both from Calabria), were used to compare the HPAEC–PAD with HPLC–RI methods. These samples were pretreated by the following procedure: 5.0 g of honey were accurately weighed dissolved in the deionized water in a 100 mL volumetric flask and the volume was properly completed. Before each analysis, all samples solutions were filtered through 0.45 mm filters (Millipore, Bedford, MA, USA).

### 2.4. HPAEC–PAD Analysis

The samples were analyzed by high-performance anion-exchange chromatography (HPAEC) with pulsed amperometric detection (PAD). The analyses were performed with a DIONEX ICS3000 system (Thermo Fischer Scientific, Germering). The analyses of sugars were conducted using a column CarboPacPA10 (4 × 250 mm) with a particle size of 10 µm preceded by a guard column CarboPacPA10 (4 × 50 mm), both purchased from Dionex Co. (Sunnyvale, CA, USA). The analyses were performed in gradient mode of milli-Q water (A) and NaOH 200 mM (B) (25%B 0 min, 25% B 0–5 min, 25–100% B 5–15 min, 100% B 15–23 min, 100–25% B 23–25 min, 25% B 25–28 min) and the flow rate was 1 mL/min. The injection volume was 20 µL. The classical pulsed amperometric detection (PAD), triple potential with a gold working electrode and a reference electrode (Ag/AgCl), was adopted as the detector. The detection mode was based on the oxidation of sugars at high pH value. However, if a single potential is applied to the electrode, oxidation products gradually poison the electrode surface, causing a loss of the analyte signal. To remove oxidation products, and therefore prevent signal loss, the electrode surface can be cleaned by a series of potentials that are applied for fixed time periods after the detection potential. In the PAD applications, the electrode is automatically cleaned and prepared for detection by each cycle of the programmed series of potentials (the waveform), thereby minimizing electrode fouling by oxidation products of the analyte and other sample compounds, and thus maintaining a consistent response. The waveform “carbohydrates (std. quad.potential)” was used. The generated current was measured and integrated with respect to time to give a net faradic charge (q) for the detection cycle. By this method, the response was measured in Coulombs/min. All experiments were conducted at 30 °C. The run time was set to 28 min and the last sugars were determined at 23 min. The chromatogram acquisition was performed with Chromeleon^®^ Chromatography Management System (Dionex Corporation). For qualitative and quantitative analysis, the external standard method was used.

### 2.5. HPLC–RI Analysis

HPLC–RI system was a Shimadzu 10A equipped with degasser DGU-14A, pumps LC-10AD, detector RID-10A and a system controller SCL-10A. The analysis was conducted using a column Waters carbohydrate (3.9 × 300 mm) with a particle size of 10 µm in conjunction with its guard column. The analyses were performed in isocratic mode. A solution of acetonitrile: milli-Q water (85:15 *v/v*) was used as eluent. The flow rate was set at 1 mL/min and the injection volume was 20 µL. The run time was set to 40 min. The performance of instruments was investigated using an external standard method.

### 2.6. Calibration Curve

Standard solutions of sugars were prepared in water by sequential dilutions. For the HPAEC–PAD analysis, the concentrations ranged from 1 to 20 mg/L for sucrose, isomaltotriose, nigerose, maltose, erlose, panose, trehalose, melibiose, maltulose, koijbiose and from 5 to 50 mg/L for melezitose, isomaltose, and palatinose. For the glucose and fructose mix, the calibration graphs were obtained using standard solutions ranging from 20 to 70 mg/L. For the HPLC–RI analysis, the standard solutions were more concentrated, ranging from 100 to 1500 mg/L for glucose and fructose mix and from 100 to 1000 for the mix composed of sucrose, isomaltotriose, nigerose, maltose, erlose, panose, melibiose, maltulose, melezitose, isomaltose, and palatinose. The amount of koijbiose and trehalose was more diluted for the HPLC–RI analysis (100 mg/L) and the signals were registered as the noise.

### 2.7. Statistical Analysis

The Unscrambler X 10.3 and XLSTAT were used to perform the statistical analysis of the data. All obtained data were statistically processed by principal component analysis (PCA) and linear discriminant analysis (LDA). The PCA is an unsupervised machine learning algorithm that facilitates the viewing data using a separation method. This is useful for data containing thousands of pieces of information (features) for each sample. When redundancy in features exists in the information, PCA reduces the dimensionality of the input data to strengthen the separation. LDA is a supervised classification technique, that is, the class member has to be known for the analysis. LDA, similar to PCA, can be considered as a dimensional reduction method to determine a lower dimension hyperplane on which the points will be projected from the higher dimension space. PCA selects a direction that retains maximal structure (variance) in a lower dimension among the data, whereas LDA selects a direction that achieves maximum separation among the given classes. The latent variable obtained in this way is a linear combination of the original variables. In the LDA method, a linear function of the variables is sought which maximizes the ratio of between-class variance and minimizes the ratio of within-class variance. Analyses of variance (ANOVA) and subsequent Tukey tests as implemented in SPSS statistical software program (IBM Statistics 20) were used to test for the significant differences in sugars profiles (*p* ≤ 0.05).

## 3. Results and Discussion

In the present study, the HPAEC–PAD combined with chemometric analysis, were used for the discrimination of monofloral honey samples, based on their botanical and geographical origin.

In accordance with other authors [27,28,29], chemometric techniques of principal component analysis (PCA) and linear discriminant analysis (LDA), were used in order to establish the most influential variables and similarities among the studied honey samples.

### 3.1. HPAEC–PAD Analysis

Seventy-four honeys samples of different botanical origin, from Calabria, Europe and China, were analyzed by the HPAEC–PAD. Thirteen different sugars were quantified in the present study consisting of two monosaccharides (glucose, fructose), seven disaccharides (trehalose, sucrose, isomaltose, palatinose, nigerose, maltose, koijbiose) and four trisaccharides (melezitose, erlose, isomaltotriose, panose).

In Figure 1, the chromatogram shows the chromatographic elution and the retention times. All 13 sugars were separated within 25 min with good resolution, except for melezitose/palatinose, which were poorly resolved under these conditions.

This method is useful to determine the minor, as well as major sugars, without the pretreatment of the sample which causes the loss of minor sugars considering the fundamental importance for honey’s discrimination. The relative abundance of monosaccharides, disaccharides, and trisaccharides in the honey samples are listed in Table 1.

Total monosaccharides in the samples were in the range of 87.8–92.2% (Table 1). According to the literature, carbohydrate composition in honey depends on different factors such as botanical and geographical origin, environmental and seasonal conditions, as well as storage and processing manipulation [11]. Fructose and glucose are the most important carbohydrates from honey and their concentrations, as well as their ratio, which have been widely used as indicators for honey origin [11,26]. The concentration of each sugar in the analyzed samples, as the mean concentration and standard deviation, are reported in Table 2. For all the investigated honey samples, reducing sugars, fructose and glucose, were found to be the major constituents, and their amounts were within the limits established by Codex Alimentarius Committee on Sugars (2001) and other studies [11,13,19,26,30,31]. In particular, fructose was the major sugar found in all honeys, followed by glucose. The mean fructose content in acacia honeys (445.46 mg/g) was higher than in other samples, chestnut (437.29 mg/g), citrus (424.61 mg/g), eucalyptus (421.14 mg/g) and sulla (398.28 mg/g). These data were in accordance with the literature especially with the results obtained in Algerian honeys [26], Croatian acacia honey [32], black locust produced in a European Atlantic area [33,34] and Serbian acacia honeys [35]. Glucose was the second major monosaccharide. The glucose content in all analyzed honeys was found in the range of 237.92–323.93 mg/g and this result was consistent with results in European honeys of Persano Oddo and Piro [36]. The content of glucose in citrus honey (323.93 mg/g) and sulla (315.46 mg/g) were higher than in other samples. In contrast, acacia (274.74 mg/g), chestnut (237.92 mg/g), eucalyptus (300.15 mg/g) and sulla (300.15 mg/g) honeys showed lowest content of glucose. The similar trend was observed for Serbian honeys where, multiflower and meadow honeys specifically contained a significant amount of glucose as compared to the other types of honey. Instead, the lowest mean value of glucose was found in the samples of acacia honey with value ranged from 22.96 g/100 g to 37.29 g/100 g [35]. In addition, Di Rosa et al. reported that among the honeys of Sicily, chestnut honeys had the lowest sugars content, with a fructose and glucose sum of 62.31 g/100 g, reflecting its characteristic bitter taste, while citrus and eucalyptus honeys showed the highest fructose content (38.08 and 38.04 g/100 g) [24]. In general, the concentration of disaccharides (trehalose, sucrose, isomaltose, palatinose, nigerose, maltose, koijbiose) in the honeys produced in Calabria ranges from 7.1% to 11.76%. The exposed data were in accordance with the previous study of Ouchemoukh et al. [26] on Algerian honeys and Mannina et al. on Sicilian honeys [22]. As shown in Table 2, chestnut and sulla honeys showed the major content of disaccharides, with a mean value of 11.76%, and 9.7%, respectively, followed by acacia, citrus and eucalyptus honeys. Maltose, palatinose, isomaltose were the major components of citrus, eucalyptus and sulla honeys. Palatinose and maltose represented the most abundant disaccharides of acacia honeys. The chestnut honeys showed a different profile with isomaltose as a principal disaccharide, followed by palatinose and maltose. The acacia honey showed a higher amount of koijbiose (6.18 mg/g) than other honeys that showed values raged from 2.52 mg/g for eucalytptus honey to 4.36 mg/g for citrus honeys. This value was in accordance with those of the lemon and orange honeys of Sicily [22] with a content of Koijbiose of 3.9 mg/g respectively. A similar trend was, also, observed for the nigerose content in sulla (6.5 mg/g) and citrus (7.9 mg/g) honeys of Calabria, compared to the same floral honeys of Sicily. However, these values were lower than chestnut and acacia, which showed 8.86 mg/g and 9.56 mg/g, respectively. The content of isomaltose was also determined and it was found in the range of 8.18 mg/g to 37.36 mg/g while the amount of maltose in the range of 11.58 mg/g to 27.27 mg/g. The high content of maltose was in accordance with the data reported by other authors for Spanish and French monofloral honeys [27,37]. Among disaccharides, the sucrose content was higher in eucalyptus and sulla honeys than other honeys, where sucrose contents ranged from 3.29% to 1.41% in citrus and acacia, respectively. The content of trisaccharides (erlose, isomaltotriose, panose) was, also, determined. As shown in Table 1, the total amount ranged between 0.46% and 2.4%. The chestnut honeys were characterized by a low content of trisaccharides compared to acacia honeys. Erlose was the most abundant trisaccharide in all samples with highest content in the acacia honeys. This oligosaccharide, produced from sucrose by the metabolism of honeybees, was also quantified in different French (Cotte et al. [37]) and Spanish unifloral honey. The content of isomaltotriose ranged from 0.16% in citrus to 1.45% in sulla honeys.

The statistical analysis of the HPAEC–PAD dataset was applied to classify the honeys produced in the Calabria Region based on the botanical origin. Due to its ability to reduce the dimensionality of multivariate data and to provide a quick preview of the data structure [28], principal component analysis allowed for the easy identification of similar groups or clusters of objects (Figure 2).

Linear discriminant analysis was also used to represent the variability of sugar data, associated with the different honey sources analyzed.

Both PCA and LDA models are valid tools to discriminate the botanical origin of honey samples (Figure 2 and Figure 3). PCA (Figure 2) shows a clear separation between all honey groups. The two principal components (PC1 and PC2) represented 66% and 24% of the total variance, respectively. The cumulative contribution rate of the first two components accounts for 90%, which represented the largest fraction of overall variability in the dataset. Similarly, LDA shows a cumulative contribution rate of the two first linear discriminant functions (DF1 and DF2) of more than 91%.

Linear discriminant analysis (LDA) biplot visualization (Figure 3) summarized the overall relations between all the variables in all the samples. The analyses of variables (Figure 3 red lines) shows that trehalose and isomaltose were determinants to classify the honeys derived from chestnut, while panose, fructose, nigerose, and erlose distinguished the acacia honeys and finally maltose and glucose, were determinants to classify the sulla honeys. Sucrose seems to be the variable determinant to classify the eucalyptus honeys.

In Appendix A, the confusion matrix (also known as an error matrix), is reported. The confusion matrix gives a representation of the statistical classification accuracy. Each row of the matrix represents the instances in a predicted class, while each column represents the instances in an actual class (or vice versa).

The correlation index (% discrimination index) is higher than 90% as shown in Appendix A. In particular, the acacia and chestnut honeys were classified with a correlation index of 100%. Considering the sulla and citrus honeys, only one sample of sulla was classified as citrus and the other way around.

According to the literature, since the citrus tree and the sulla species bloom in the same period, the derived honey can often be unidentified.

### 3.2. Comparison HPAEC–PAD Method vs. Official Method HPLC–RI

The developed HPAEC–PAD method to classify the honey samples, based on botanical and geographical origin, using their sugar profile was, also compared with the official HPLC–RI method, (Official Gazzetta n°185 of 11 August 2003). Using the sugar solutions and three different honeys, the analytical performance of both methods was evaluated. The HPLC–RI technique showed a lower capacity of separating and quantifying sugars compared with HPAEC–PAD. Applied to the honey samples, both methods showed similar amounts only for glucose and fructose (Table 3). The performance of the methods according to range, linearity, limit of detection, and the limit of quantification both for HPAEC–PAD and for HPLC–RI (Table 4 and Table 5) was evaluated. The instrumental responses were, also evaluated by the calibration curves with respect to the concentration of the analytes in the matrices under study in the working range. The precision of injection was demonstrated replicating the injections of the standard solution at a different concentration, 10 mg/L for HPAEC–PAD and 500 mg/L for HPLC–RI. The calibration curves were obtained by plotting the concentration of each standard against their detector signal expressed as the area under the chromatographic curve. The correlation coefficients (r^2^) of detector response vs. standard concentration were greater than 0.998 for all standards analyzed using both RI and PAD.

The detection limit was expressed as LOD = 3.3 σ/S and the limit of quantification was expressed as LOQ = 10 σ/S. σ is the standard deviation of the response, and S is the slope of the calibration curve. The slope S may be estimated from the calibration curve of the analyte. The residual standard deviation of a regression line or the standard deviation of y-intercepts of regression lines were used as the standard deviation.

### 3.3. Comparison Acacia Honeys from Calabria vs. Acacia Honeys from Europe and China

The potential of the proposed HPAEC–PAD method coupled with the chemometric analysis was tested by applying it to honeys from China, and Europe, in comparison with those of Calabria (Table 6). For this purpose, only acacia honeys were used. Samples from China showed the highest total amount of sugars (average value 839.61 mg/g) compared to both the European and Calabrian acacia honeys. According to Schievano et al. [38], they were characterized by a high content of monosaccharides (799.31 mg/g). In addition, the values of disaccharides and trisaccharides, respectively, of 39.93 mg/g and 1.37 mg/g, were higher than those of the acacia honeys from China found by Schievano [38] but lower than those of the honey from Europe and Calabria. European acacia honeys showed a content of disaccharides and trisaccharides of 77.396 and 18.10 mg/g, respectively, whereas the acacia honeys of Calabria showed a mean content of disaccharides of 67.3 mg/g and trisaccharides of 19.7 mg/g.

Figure 4 shows that principal component analysis (PCA) allowed easy identification, grouping the acacia honey in clusters of different geographical origins.

Additionally, Figure 5 shows that the linear discriminant analysis (LDA), was able to build a classification model, which can be used to predict the unknown samples (Appendix A).

The analyses of the loading plot (Figure 5, red lines) show that trehalose, palatinose, sucrose, and isomaltose were determined to classify the acacia honeys from Europe, while fructose, nigerose, and maltose to distinguish the samples from Calabria, and only the glucose amount to classify the China samples.

## 4. Conclusions

An instrumental technique based on high-performance anion exchange chromatography with pulsed amperometric detection (HPAEC–PAD) method combined with chemometric analysis was developed to investigate the sugar *fingerprints* in honey. For the first time, the developed method was also applied to honeys of different botanical origin produced in Southern Italy (Calabria Region).

The developed HPAEC–PAD method compared with the Italian official HPLC–RI method showed a greater capacity for separating and quantifying sugars.

Principal component and linear discriminant analyses were used to describe the sugar variability of HPAEC–PAD data associated with the different honey sources, and the possibility of relating these values to either floral or geographical differentiation.

Monosaccharides, disaccharides, trisaccharides, were determined in honey samples with a different botanical, within the Calabria Region, and geographical origin by comparison of European and Chinese honeys. All studied honeys meet the criteria set by the Council Directive 2001/110/EC and the Revised Codex Standard for Honey, regarding the sugar content. Fructose and glucose were the most abundant reducing sugars, in agreement with the literature data. The monosaccharide and oligosaccharide profile of analyzed honeys was useful to discriminate the sample according to their different floral characteristics.

The fingerprint of acacia honeys is characterized by the highest fructose content, while palatinose and maltose and isomaltose were the main disaccharides. The high erlose content makes this sugar an excellent candidate as a marker determined in the discrimination of acacia honeys compared to other floral qualities.

Citrus honeys were characterized by the highest content of maltose and by the high content of palatinose and isomaltose. The fingerprint of chestnut honeys was characterized by the highest content of disaccharides of which isomaltose was the main one. This sugar, with trehalose, are potential candidates as chestnut markers, although palatinose and maltose also contribute to the fingerprint. Finally, among minor sugars, panose, nigerose, erlose and maltose were decisive for classifying the sulla honeys.

European acacia honeys, including those produced in Calabria, were characterized by a higher content of disaccharides and trisaccharides than Chinese ones, which had the highest content of monosaccharides. Sucrose, the trisaccharides trehalose and palatinose, and isomaltose were determinants to classify the acacia honeys from Europe, while fructose, nigerose, and maltose to discriminate the samples from Calabria.

The described model confirmed that some sugars can be used as markers of authenticity, allowing the botanical and geographical discrimination of honeys. The analytical approach combined with chemometric analysis allows to assimilate minor sugars, disaccharides and trisaccharides, to markers. Therefore, in the future, further investigations will be conducted on a large sampling of different unifloral and multifloral honeys produced both in Calabria and in other EU and non-EU countries.

## Figures and Tables

**Figure 1 foods-09-01625-f001:**
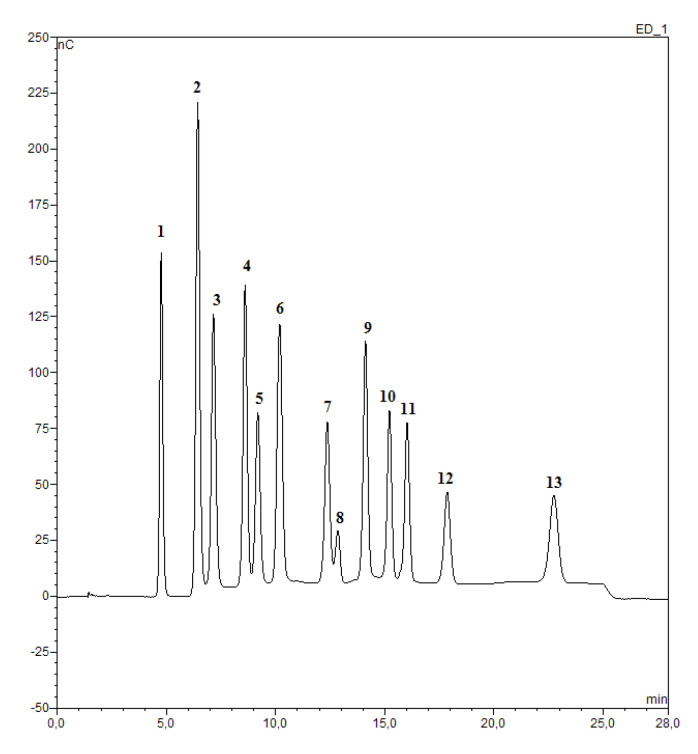
Chromatograms obtained by the high-performance anion-exchange chromatography with pulsed amperometric detection (HPAEC–PAD) analysis of a standard mix of thirteen sugars (1) threalose, (2) glucose, (3) fructose, (4) sucrose, (5) isomaltose, (6) koijbiose, (7) melezitose, (8) palatinose, (9) isomaltotriose, (10) nigerose, (11) maltose, (12) erlose and (13) panose.

**Figure 2 foods-09-01625-f002:**
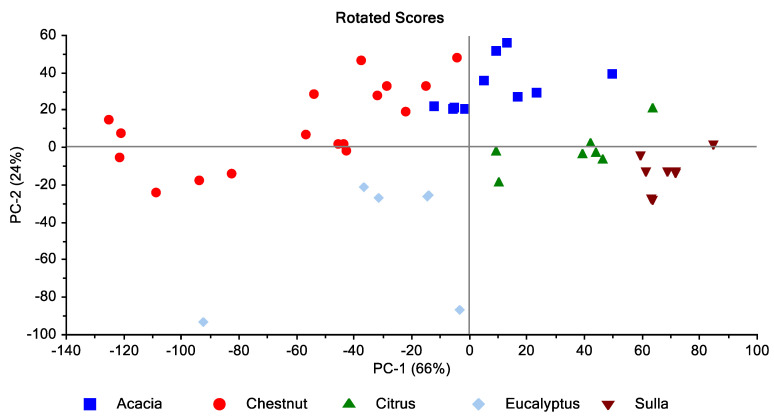
The principal component analysis (PCA) of honeys of Calabria classified on botanical origin.

**Figure 3 foods-09-01625-f003:**
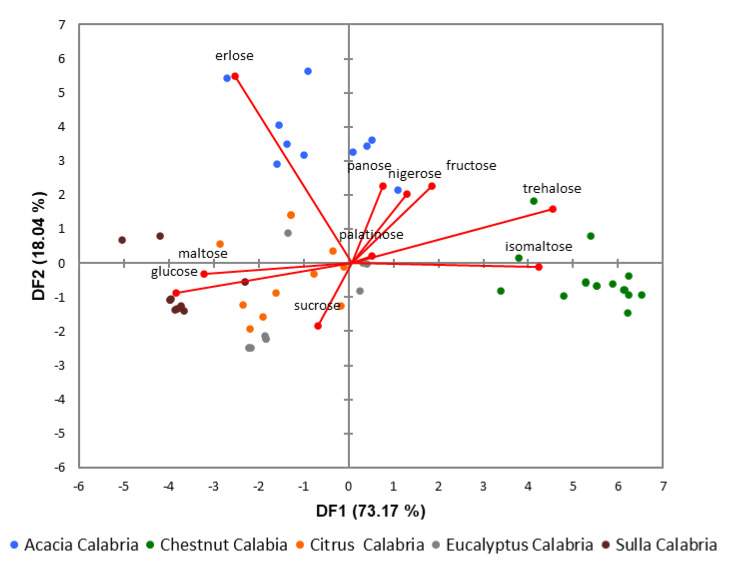
Linear discriminant analysis (LDA) biplot of honeys classified based on botanical origin.

**Figure 4 foods-09-01625-f004:**
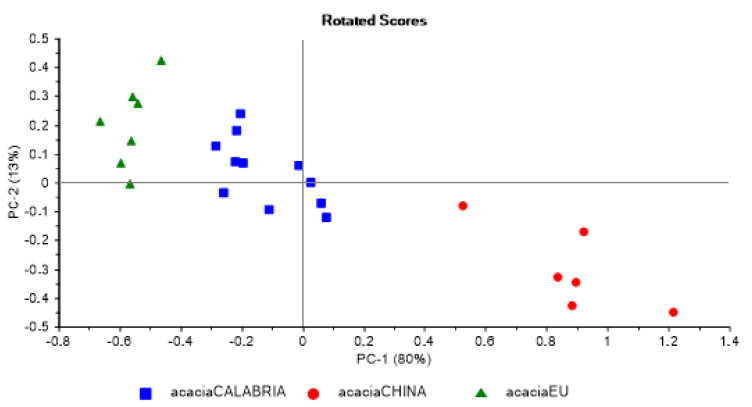
PCA of acacia honeys classified based on geographical origin.

**Figure 5 foods-09-01625-f005:**
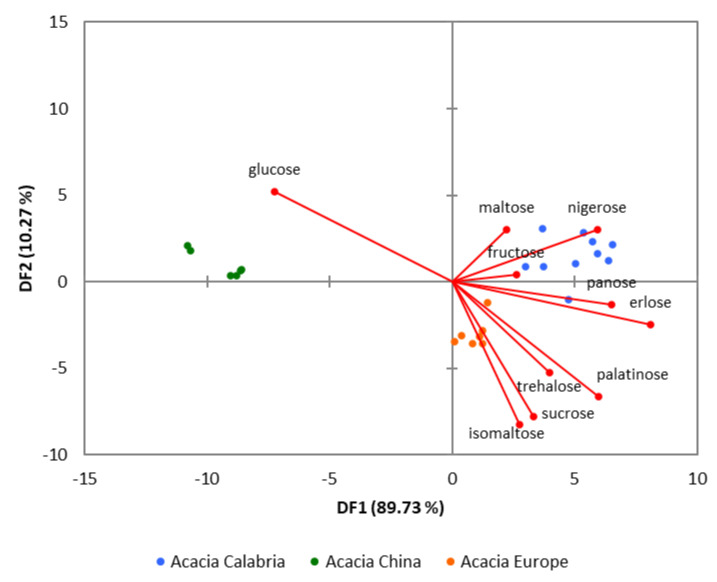
LDA biplot of acacia honeys classified based on their geographical origin.

**Table 1 foods-09-01625-t001:** Relative abundance (%) of monosaccharides (MS), disaccharides (DS), and trisaccharides (TS) in honey samples from Calabria.

Sugars (%)	Honey Type (Number of Samples)				
	Citrus	Eucalyptus	Sulla	Chestnut	Acacia
	(*n* = 12)	(*n* = 10)	(*n* = 12)	(*n* = 16)	(*n* = 10)
Total MS					
Mean ± sd	91 ^b,c^ ± 1,4	92.2 ^c^±1.15	89.0 ^a,b^ ± 3.8	87.77 ^a^ ± 2.67	89.2 ^a,b^ ± 0.94
Range	88.3–92.6	90.1–93.3	77.7 ± 92.6	84.1–91.1	88.4–90.7
Total DS					
Mean ± sd	8.0 ^a^ ± 1.3	7.1 ^a^ ± 0.98	9.7 ^a,b^ ± 4.0	11.76 ^c^ ± 2.67	8.3 ^a^ ± 0.64
Range	6.9–10.8	6.1–8.5	6.2–21.6	8.8–15.7	7.8–9.7
Total TS					
Mean ± sd	1.1 ^b^ ± 0.5	0.8 ^a,b^ ± 0.26	1.2 ^b^ ± 0.4	0.46 ^a^ ± 0.41	2.4 ^c^ ± 0.55
Range	0.5–2	0.6–1.4	0.8–1.2	0.2–1.2	1.4–3.3

Results are expressed as the mean values ± standard deviations. ^a^, ^b^ and ^c^ letters show significant differences among the honey type—Tukey’s test, *p*-level ≤ 0.05.

**Table 2 foods-09-01625-t002:** Sugars profile in the different honeys of Calabria.

Sugars (MG/G)	Honey Type (Number of Samples)				
	Citrus	Eucalyptus	Sulla	Chestnut	Acacia
	(*n* = 12)	(*n* = 10)	(*n* = 12)	(*n* = 16)	(*n* = 10)
Glucose					
Mean ± sd	323.93 ^c^ *±* 32.32	300.15 ^b.c^ *±* 27.94	315.46 ^b.c^ *±* 53.39	237.92 ^a^ *±* 26.19	274.74 ^b^ *±* 25.59
Range	270.38–366.89	277.22–332.60	190.30–354.42	188.93–278.58	237.04–305.84
Fructose					
Mean ± sd	424.61 ^a.b^ *±* 13.93	421.14 ^a.b^ *±* 57.86	398.28 ^a^ *±* 22.31	437.29 ^b^ *±* 21.91	445.46 ^b^ *±* 27.77
Range	405.67–447.07	338.21–464.51	332.20–420.82	407.92–473.61	404.20–480.96
Trehalose					
Mean ± sd	2.43 ^b^ *±* 0.56	1.58 ^a^ *±* 0.77	1.52 ^a^ *±* 0.82	4.17 ^c^ *±* 0.61	2.89 ^b^ *±* 0.42
Range	1.34–3.21	0.69–2.40	0.82–3.71	3.41–5.07	2.51–3.65
Sucrose					
Mean ± sd	2.41 ^a^ *±* 3.05	7.34 ^a^ *±* 10.30	6.82 ^a^ *±* 9.65	3.29 ^a^ *±* 2.67	1.41 ^a^ *±* 0.94
Range	0.12–9.60	0.08–22.25	0.08–22.39	0.34–7.10	0.38–2.76
Isomaltose					
Mean ± sd	10.87 ^a^ *±* 1.80	8.18 ^a^ *±* 3.66	14.00 ^a^ *±* 8.99	37.36 ^b^ *±* 13.69	15.46 ^a^ *±* 2.69
Range	8.25–14.68	3.72–13.09	6.61–38.46	22.14–57.63	9.96–17.40
Koijbiose					
Mean ± sd	4.36 ^b^ *±* 1.28	2.52 ^a^ *±* 1.28	4.36 ^b^ *±* 1.28	4.00 ^b^ *±* 0.94	6.18 ^c^ *±* 1.33
Range	2..95–5.26	nd–5.84	2..95–5.26	nd–6.21	4.71–8.30
Palatinose					
Mean ± sd	15.40 ^a.b^ *±* 4.24	13.56 ^a^ *±* 5.07	19.37 ^b^ *±* 9.68	21.68 ^a.b^ *±* 4.17	19.01 ^a.b^ *±* 3.02
Range	12.19–19.95	6.82–18.24	10.79–33.52	13.52–28.42	15.57–26.35
Isomaltotriose					
Mean ± sd	0.16 ^a^ *±* 0.06	0.81 ^b^ *±* 0.23	1.45 ^c^ *±* 0.192	1.25 ^b.c^ *±* 0.192	0.31 ^a^ *±* 0.16
Range	nd–0.23	0.4–1.06	nd–1.96	nd–1.36	nd–0.43
Nigerose					
Mean ± sd	7.09 ^a^ *±* 3.60	5.31 ^a^ *±* 3.09	6.49 ^a^ *±* 4.74	8.86 ^a^ *±* 3.93	9.56 ^a^ *±* 4.20
Range	2.53–14.51	2.98–10.20	2.53–16.35	5.46–20.01	3.27–14.93
Maltose					
Mean ± sd	27.27 ^c^ *±* 7.08	18.64 ^a.b^ *±* 6.30	23.56 ^b.c^ *±* 7.37	11.58 ^a^ *±* 3.94	18.94 ^b^ *±* 6.90
Range	20.27–42.07	9.96–32.66	10.51–33.02	5.76–20.68	12.59–31.53
Erlose					
Mean ± sd	7.64 ^b^ *±* 3.20	5.25 ^a.b^ *±* 2.07	8.36 ^b^ *±* 4.24	1.71 ^a^ *±* 2.35	17.68 ^c^ *±* 4.39
Range	3.70–13.90	3.51–10.21	Nd–16.83	nd–8.34	10.56–24.72
Panose					
Mean ± sd	1.15 ^a.b^ *±* 0.94	0.75 ^a^ *±* 0.61	1.65 ^a.b^ *±* 1.35	1.72 ^a.b^ *±* 0.74	2.06 ^b^ *±* 0.87
Range	0.17–2.96	Nd–1.48	0.37–5.09	0.98– 3.22	1.42–4.23

Results are expressed as the mean values ± standard deviations. ^a^, ^b^ and ^c^ letters show significant differences among the honey type—Tukey’s test, *p*-level ≤ 0.05.

**Table 3 foods-09-01625-t003:** Comparison of the amount of fructose and glucose in two citrus honey and one chestnut obtained using HPAEC–PAD and HPLC–RI methods.

	Fructose (mg/g)		Glucose (mg/g)	
	HPLC–RI	HPAEC–PAD	HPLC–RI	HPAEC–PAD
Citrus Honey 1	448.56 ± 12.71	411.84 ± 3.27	320.47 ± 11.47	355.16 ± 2.53
Citrus Honey 2	421.87 ± 12.55	417.12 ± 2.45	325.23 ± 13.09	339.48 ± 3.35
Chestnut Honey	473.32 ± 13. 27	471.16 ± 2.39	232.50 ± 15. 27	253.04 ± 1.33

**Table 4 foods-09-01625-t004:** Evaluation of the linearity range, correlation factor, calibration curves, LOD, LOQ, and precision (expressed as relative standard deviation, RSD%) of the analyzed standards by HPLC–RI.

	Range (mg/L)	Rq	Calibration Curve	LOD (mg/L)	LOQ (mg/L)	RSD %
Fructose	150–1500	1	y = 145.21x + 8836.5	42.04	127.40	21.73
Glucose	150–1500	1	y = 172.68x +14766.5	57.95	175.61	28.73
Isomaltose	250–1000	0.998	y = 126.38x + 4832	52.33	158.57	85.28
Panose	250–1000	1	y = 173.26x − 4793	37.32	113.09	120.04
Isomaltotriose	250–1000	0.998	y = 137x − 2329	42.65	129.25	98.13

**Table 5 foods-09-01625-t005:** Evaluation of linearity range, correlation factor, calibration curves, LOD, LOQ, and precision (expressed as relative standard deviation, RSD%) of the analyzed standards by HPAEC–PAD.

	Range (mg/L)	RQ	Calibration Curve	LOD (mg/L)	LOQ (mg/L)	RSD %
Fructose	10–50	1	y = 2.85x + 5.84	0.86	2.61	0.17
Glucose	10–50	1	y = 4.25x+ 0.138	0.24	0.74	0.08
Isomaltose	5–50	1	y = 2.79x +0.80	0.17	0.52	0.12
Panose	1–20	1	y = 1.72x – 0.06	0.17	0.52	0.07
Isomaltotriose	1–20	1	y = 2.55x-0.18	0.24	0.74	0.19
Sucrose	1–20	1	y = 1.94x – 0.25	0.2	0.63	0.04
Erlose	1–20	1	y = 1.29x -0.07	0.14	0.41	0.06
Maltose	1–20	1	y = 1.70x + 0.19	0.076	0.23	0.48
Nigerose	1–20	1	y = 1.65x+ 0.23	0.07	0.21	0.35
Melezitose	1–50	1	y = 1.95x + 0.069	0.15	0.48	0.37
Trehalose	1–20	1	y = 2.64x+ 0.22	0.05	0.16	0.05
Koijbiose	1–20	1	y = 2.22x + 0.39	0.18	0.53	0.47
Palatinose	5–50	1	y = 1.05x + 0.89	0.38	1.15	0.80

**Table 6 foods-09-01625-t006:** Sugars amount in the acacia honeys of Calabria vs. China and Europe.

Sugars (MG/G)	Acacia Honey (Number of Samples)		
	Calabria	Europe	China
	(*n* = 10)	(*n* = 7)	(*n* = 6)
Total			
Mean ± sd	807.2 ^a^ ± 53.0	791.02 ^a^ ± 24.8	839.61 ^a^ ± 12.1
Range	752.64–863.91	90.1–93.3	824.74 ± 859.01
Total MS			
Mean ± sd	720.2 ^a^ ± 49.8	695.52 ^a^ ± 28.8	799.31 ^b^ ± 11.6
Range	664.07–776.69	671.19–745.41	787.95 ± 807.70
Total DS			
Mean ± sd	67.3 ^a^ ± 5,2	77.39 ^b^ ± 12.9	38.93 ^b^ ± 11.02
Range	62.60–74.47	61.28–87.34	25.05–48.37
Total TS			
Mean ± sd	19.7 ^a^ ± 4.8	18,10 ^b^ ± 1,8	1.37 ^b^ ± 0.34
Range	15.95–26.22	16.09–21.37	1.09–1.84

Results are expressed as the mean values ± standard deviations. ^a^, ^b^ and ^c^ letters show significant differences among the honey types—Tukey’s test, *p*-level ≤ 0.05.

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
