# Peer review of "High-Performance Anion Exchange Chromatography with Pulsed Amperometric Detection (HPAEC–PAD) and Chemometrics for Geographical and Floral Authentication of Honeys from Southern Italy (Calabria region)"

_foods, 2020, doi:10.3390/foods9111625_

Round 1
Reviewer 1 Report
1.Dear Dr. Sonia Carbetta and coauthors, your research article:"High-performance anion exchange chromatography with pulsed amperometric detection (HPAEC-PAD) and chemometrics for geographical and floral authetication of honeys from Southern Italy (Calabria's region) was reviewed.
2. It is requested to correct Figure 2: Unclear data. The picture should be redrown with smaller fonts with the numbers instead the names of the honeys!!
3. It is requested to correct Figure 4: Unclear data. The picture should be redrown with smaller fonts with the numbers instead the names of the honeys!!
4. You should correct:
Line No.: Requested correction:
23 Del. 74, Add: Seventy - four
29 Add: developed, between
55 Add: ,
70 Del.: have been; Add: were
71 Add: the
72 Del.: have been; Add: were
75 Del.: have; Add: were
84 Del: which determines; Add: determining
86 Del.: include; Add: including the
88 Add: and
95 Del.: has been; Add: were
99 Del.: indicator; Add: factor
101 Add: the
103 Add: the
111 Del.: have been; Add: were
116 Del.: have been; Add: were
Del: compared; Add: in comparison
138 Del.: have been; Add: were
139 Del.: have been; Add: were
140 Add: the
160 Add: the
211 Del.: , Add: and
235 Add: ,
240 Del.: have shown; Add: showed
242 Del.: have been; Add: were
250 Del.: have been; Add: were
Del.: compared; Add: in comparison
281 Figure 2: Unclear data. The picture should be redrown with the smaller
fonts with numbers instead of the names of the honeys.
283 Add: to represent
288 Add: between
291 Del.: represented; Add: represent
292 Del.: larges; Add: largest
298 Add: ,
306 Add: the
312 Del.: has been; Add: was
316 Add: the
317 Del.: is ; Add: are
322 Add: and
341 Figure 4: Unclear picture!! Should be redrawed with numbers instead of
the names of the honeys
366 Del.: replicate; Add: replicating the
367 Add: a
388 Add: described
Del.: was able to; Add: can
392 Add: ,

Author Response
Dear Editor and Reviewers,
Thank you for your comments and suggestions on our manuscript entitled: “High-performance anion exchange chromatography with pulsed amperometric detection (HPAEC-PAD) and chemometrics for geographical and floral authentication of honeys from Southern Italy (Calabria region)”
We are grateful for the opportunity to answer the questions related to our study submitted to the Foods. The manuscript was modified (written in red) according with the reviewers suggestions. The corrections or specific answers are listed below point by point.
- It is requested to correct Figure 2: Unclear data. The picture should be redrown with smaller fonts with the numbers instead the names of the honeys!!
- It is requested to correct Figure 4: Unclear data. The picture should be redrown with smaller fonts with the numbers instead the names of the honeys!!
Authors: The pictures were modified the names of samples was deleted and the legend was added
- You should correct:
Line No.: Requested correction:
23 Del. 74, Add: Seventy – four
Authors: Modified
29 Add: developed, between
Authors: Added
55 Add: ,
Authors: Added
70 Del.: have been; Add: were
Authors: Deleted have been and added were
71 Add: the
Authors: Added
72 Del.: have been; Add: were
Authors: Deleted have been and added were
75 Del.: have; Add: were
Authors: deleted have been and added were
84 Del: which determines; Add: determining
Authors: modifed
86 Del.: include; Add: including the
Authors: modified
88 Add: and
Authors: added
95 Del.: has been; Add: were
Authors: deleted have been and added were
99 Del.: indicator; Add: factor
Authors: modified
101 Add: the
Authors: added
103 Add: the
Authors: added
111 Del.: have been; Add: were
Authors: deleted have been and added were
116 Del.: have been; Add: were
Del: compared; Add: in comparison
Authors: both modified
138 Del.: have been; Add: were
Authors: deleted have been and added were
139 Del.: have been; Add: were
Authors: deleted have been and added were
140 Add: the
Authors: added
160 Add: the
Authors: added
211 Del.: , Add: and
Authors: modified
235 Add: ,
Authors: added
240 Del.: have shown; Add: showed
Authors: modified
242 Del.: have been; Add: were
Authors: deleted have been and added were
250 Del.: have been; Add: were
Del.: compared; Add: in comparison
Authors: both modified
281 Figure 2: Unclear data. The picture should be redrown with the smaller fonts with numbers instead of the names of the honeys.
283 Add: to represent
Authors: added
288 Add: between
Authors: added
291 Del.: represented; Add: represent
Authors: modified
292 Del.: larges; Add: largest
Authors: corrected
298 Add: ,
Authors: added
306 Add: the
Authors: added
312 Del.: has been; Add: was
Authors: deleted have been and added were
316 Add: the
Authors: added
317 Del.: is ; Add: are
Authors: corrected
322 Add: and
Authors: added
341 Figure 4: Unclear picture!! Should be redrawed with numbers instead of the names of the honeys
366 Del.: replicate; Add: replicating the
Authors: modified
367 Add: a
Authors: added
388 Add: described
Del.: was able to; Add: can
Authors: both modified
392 Add: ,
Authors: added
Reviewer 2 Report
The article presents the sugars profiling of different botanical origin honeys from Southern Italy,Calabria Region.The samples were compared with honey from Europe and China and have been analyzed by High performance anion-exchange chromatography with pulsed amperometric detection. In the article the variability of carbohydrate data associated with the different honey profiles was discussed by chemometrics methods. Based on the obtained results, the analytical advantages of the HPAEC-PAD method have been demonstrated comparedto the HPLC-RI Italian official method.
The article could be interesting and it needs some improvements, listed below:
- The Authors should give more details about the limits established by Codex Alimentarius Committee (line 47-49, page 2).
- All results are presented with mean values. In my opinion, it is not sufficient. Please, assign the results into homogeneous groups.
- In the chapter "Results and discussion" there is too little discussion with the results obtained by other Authors. Please complete the chapter with disussion.
- Please, correct the mean values in table 6.
- Chapter "Conclusions" should be corrected and rewritten. Tables' numbers should not be given in conclusions. The chapter should be written more carefully and completed with more information about the results obtained. Its content is insufficient.
Author Response
Dear Editor and Reviewers,
Thank you for your comments and suggestions on our manuscript entitled: “High-performance anion exchange chromatography with pulsed amperometric detection (HPAEC-PAD) and chemometrics for geographical and floral authentication of honeys from Southern Italy (Calabria region)”
We are grateful for the opportunity to answer the questions related to our study submitted to the Foods. The manuscript was modified (written in red) according with the reviewers suggestions. The corrections or specific answers are listed below point by point.
- The Authors should give more details about the limits established by Codex Alimentarius Committee (line 47-49, page 2).
Authors: More details about the limits established by Codex Alimentarius Committee, relating sugars, were added.
- All results are presented with mean values. In my opinion, it is not sufficient. Please, assign the results into homogeneous groups.
Authors: The Anova analysis was added for all tables
- In the chapter "Results and discussion" there is too little discussion with the results obtained by other Authors. Please complete the chapter with disussion.
Authors: More discussion with the results obtained by other Authors were added
- Please, correct the mean values in table 6.
Authors: The mean values were corrected
- Chapter "Conclusions" should be corrected and rewritten. Tables' numbers should not be given in conclusions. The chapter should be written more carefully and completed with more information about the results obtained. Its content is insufficient.
Authors: The paragraph was rewritten adding more information and the tables’ number were deleted